# Successful Management Foreign Body Aspiration Associated with Severe Respiratory Distress and Subcutaneous Emphysema: Case Report and Literature Review

**DOI:** 10.3390/medicina58030396

**Published:** 2022-03-07

**Authors:** José Juan Gómez-Ramos, Alejandro Marín-Medina, Alexandro Azael Castillo-Cobian, Oscar Gabriel Felipe-Diego

**Affiliations:** 1Departamento de Urgencias, Hospital General de Zona No. 89 IMSS, Guadalajara 44190, Jalisco, Mexico; josejuan79@yahoo.com (J.J.G.-R.); aza9402@gmail.com (A.A.C.-C.); osgafedi@gmail.com (O.G.F.-D.); 2Departamento de Biología Molecular y Genómica, Centro Universitario de Ciencias de la Salud (CUCS), Universidad de Guadalajara (UdeG), Guadalajara 44100, Jalisco, Mexico; 3Centro Universitario de Ciencias de la Salud (CUCS), Universidad de Guadalajara (UdeG), Guadalajara 44100, Jalisco, Mexico

**Keywords:** foreign bodies, bronchoscopy, airway obstruction, subcutaneous emphysema, tracheostomy

## Abstract

The presence of a foreign body in the airway is a potentially life-threatening clinical condition that requires urgent medical attention. We present a case of a 12-year-old boy who presented in the emergency room with a history of an episode of choking after aspiration of a foreign body, followed by severe respiratory distress and subcutaneous emphysema. Chest radiography revealed hyperinflation data, pneumothorax, and subcutaneous emphysema data. The flexible bronchoscope examination showed the presence of an inorganic foreign body impacted on the carina with tracheal lesions and laryngeal edema. It was necessary to perform a tracheostomy for its definitive extraction. The gold standard in the treatment of foreign body aspiration is bronchoscopy; although, in children, the technique adopted continues to be controversial, flexible bronchoscopy can be effective and very useful.

## 1. Introduction

The presence of respiratory distress in children is one of the main reasons for consultation in the emergency room (ER), responsible for up to 10% of all consultations [1]. A significant proportion of respiratory distress cases in children are caused by the presence of a foreign body in the airway (FBa).

Although most cases occur in the population under 5 years of age (60–80%), around 15% of cases occur in the population aged 5–15 years, and approximately 6% in those over 15 years of age [2].

According to reports from the US Centers for Disease Control and Prevention, for the year 2000, the incidence rate of FBa was 29.9/100,000 children, responsible for 160 deaths [3]. However, the existing data, in relation to mortality due to the aspiration of a foreign body (FB) in pediatric age, are very few and varied, basically depending on the age group, the socioeconomic environment, and the clinical scenario studied. In this sense, Montana et al. [4] report an extra-hospital mortality of 36.4% related, above all, to those fatal cases where the foreign body became lodged in the larynx or trachea [5]. Meanwhile, in-hospital mortality is between 0.26 and 13.6%, and is mainly associated with late hypoxia complications and bronchoscopy complications [4], with a downward trend over time most likely related to advanced modern bronchoscopic techniques.

Bronchoscopy is the diagnostic and therapeutic method indicated when FB inhalation is suspected, especially in children. It is considered the gold standard in the identification and localization of FBa [6]. The evidence shows that morbidity caused by bronchoscopy is lower than that caused by an undiagnosed FB, with a low risk of complications if performed in the first 24 h [6]. The diagnosis is clinical, and a high suspicion should always be maintained in patients presenting with a cough, shortness of breath, cyanosis, unilaterally diminished breath sounds, and air trapping on chest X-ray [7]. 

We present a case of an adolescent with an FB aspiration and review the literature regarding clinical manifestations, surgical and radiological findings, as well as the management of patients with this condition.

## 2. Case Report

A 12-year-old boy was admitted to the ER of a secondary care hospital with severe respiratory distress. After leaving school, he ingested a sweet and sour chamoy-flavored liquid candy in a container with an atomizer cap. He suddenly began to cough violently with an impending choking sensation. Immediately, he presented difficulty in breathing and FB sensation. He was taken to the emergency department (ED) by an ambulance for his respiratory distress.

Following his admission to the ER, the presence of an FBa was suspected. The choking episode, with the subsequent respiratory distress, was witnessed by his classmates and reported to the emergency services. 

Upon his arrival, he was anxious, diaphoretic, with an evident increase in breathing work, soporous, and had difficulty articulating words. An intravenous line had been placed, in addition to supplemental oxygen, through a reservoir mask. 

On physical examination, he had a blood pressure of 130/107 mmHg, respiratory rate of 28 bpm, heart rate of 118 bpm, a temperature of 36 °C, and an oxygen saturation of 99% with supplemental oxygen through a mask with a reservoir. He presented a poor general condition, with pale skin and mucous membranes. There was swelling in his neck and right mammary region, with crepitus on palpation, and evident thoracic asymmetry. His respiratory movements were diminished. In auscultation, the vesicular murmur was audible; although it greatly diminished in its intensity during inspiration, it was not audible on exhalation. 

Once in the ER, a complete blood count, and blood chemistry, coagulation, and arterial blood gases tests were performed, as well as a simple chest X-ray in anteroposterior projection in a sitting position. Given the clinical status of the patient and the severity of the condition, it was not possible to perform other imaging studies, such as a lateral projection chest X-ray or computed tomography of the chest. 

The arterial gases showed decompensated respiratory acidosis with hypercapnia. The chest X-ray (Figure 1) showed no evidence of an FB; however, it presented frank hyperinflation data, a small pneumothorax located in the upper part of the left hemithorax, as well as subcutaneous emphysema in the neck and chest. The rest of the laboratory studies did not present relevant data for the case.

Given the absence of a pediatric pulmonology service and the impossibility of performing a bronchoscopy in our hospital, he was referred to the pediatric pulmonology service of a tertiary care hospital (our reference hospital), due to the suspicion of an FBa. The performance of flexible bronchoscopy under general anesthesia was considered as the appropriate diagnostic–therapeutic method (since a rigid bronchoscope was not available), and was carried out two hours after their arrival (and five hours after the onset of the clinical picture).

The bronchoscopy examination showed abundant blood secretions in the oral cavity (most likely of tracheal origin), as well as in the supraglottic space, with epiglottis in omega, and cartilage without alterations. It showed the glottic space with vocal cords without alterations, the subglottic space with edematous stenosis with an approximately 30% reduction of light, and the trachea with membranous and cartilaginous parts with hyperemic mucosa and blood dotting (Figure 2). An inorganic FB was seen in the main carina; however, it was not possible to extract it, as it was trapped in the subglottic space, and required an emergency tracheotomy which was performed without complications with FB extraction through a Björk flap.

The patient was admitted to the pediatric intensive care unit (PICU), where he stayed for a period of three days. He was discharged six days later.

## 3. Discussion

The presence of an FBa usually occurs at any age; some studies estimate an incidence of 0.66 per 100,000 inhabitants [2]. Although it can occur in adulthood, it is more common in the pediatric age group, especially in children between 1 to 3 years of age, and up to 16% occurs in the ages between 5 to 15 years [2,3].

Some factors related to the risk of presenting an FBa involve mainly those related to specific anatomical conditions of the pediatric age group and age-specific behavior. Accidental aspiration during crying, laughing, or playing is common, although less frequent [6].

For descriptive purposes, the foreign bodies (FBs) have been divided into two large groups according to their nature: organic and inorganic. Multiple studies show a higher frequency of FBs with organic composition, the latter being responsible for more severe inflammatory reactions [8].

The location of the FB usually depends mainly on the size of the FB and the age of the victim. Findings have been reported in virtually the entire respiratory tract [3,6,9]. Most case series reported coincide, positioning the right main bronchus as the most frequent site, and the left main bronchus as the second [7,8,9,10]. As in the case we present, some cases series mention the carina as a localization site with variable frequencies ranging from 5.4% to 29%, emphasizing that the impact of an FB on this site are usually fatal cases [9,10,11].

Four types of obstruction have also been described (Table 1) and according to the clinical manifestations shown by our patient, the type of obstruction presented was a type II or check valve [12]. This caused air trapping in the alveolar spaces with gradual increase in interalveolar pressure, causing its rupture [13]. The escaped air dissected the pulmonary muscular sheath causing interstitial emphysema, and later subcutaneous emphysema, in the thorax that extended to the neck through the cervical fascia, which may have caused pneumomediastinum [14].

In general, physical examination can often find tachypnea, stridor, decreased breath sounds, wheezing and/or crepitus, and in some cases, fever. Our patient entered the ER with tachypnea and with a significant decrease in respiratory sounds [7,10]. In addition to the signs already described, some studies mention the possibility of decreased air intake in auscultation, abnormal breath sounds, asymmetry of chest inspection, nasal flaring, abnormal respiratory wheezing, nose pain, rhonchi, use of accessory muscles of respiration, purulent discharge, bradypnea, hypoxia, and hypercapnia [2,8].

High rates of complications have been reported, mainly associated with the establishment of a late diagnosis. Some of the reported complications include unilateral hyperinflation, mediastinal shift, laryngeal edema, tracheal laceration, empyema, pulmonary edema, atelectasis, persistent fever, bronchiolitis, pneumonia, presence of subcutaneous emphysema, pneumomediastinum, and pneumothorax. [4,8,11]. Subcutaneous emphysema was one of the clinical signs present in our patient and is considered a rare presentation in cases of FBs in the airway [13,15]. Foltran et al. [14] report only five cases in two different publications, estimating a frequency of 1.3% [13]; our patient presented grade IV subcutaneous emphysema, which included the entire thoracic wall and neck.

The diagnosis is usually clinical, although sometimes it can be a challenge. It was found that the presence of focal hyperinflation on the chest X-ray, the asphyxia crisis witnessed, and a leukocyte count greater than 10,000 /mL show a cumulative proportion of up to 100% when all three are present [6,7]. 

On suspicion of the aspiration of an FB, chest radiography is suggested, with anteroposterior and lateral vertical projections; a lateral soft tissue neck radiograph is also suggested [1,6]. The most frequent findings are the visualization of the FB, although it is considered that of the FBs aspirated only 8.2–24% are radiopaque, lobar or segmental radiolucency, areas of atelectasis, and inflammatory consolidation of the pulmonary parenchyma unilateral or bilateral hyperinflation. The presence of pneumothorax, subcutaneous emphysema, and pneumomediastinum are less frequent [1,12,14]. Although it is also common to routinely obtain a simple lateral decubitus chest X-ray, some studies confer it a limited role in diagnosis, with only a sensitivity of 27% and a specificity of 67%. On the other hand, although computed tomography is superior to a chest X-ray (especially in the case of radiopaque foreign bodies), with a sensitivity of 100% and a specificity of 66.7%, it has some limitations, such as: radiation exposure and the restriction of movement required for high-quality scans, which is often not feasible in people with respiratory distress [2]. In the presence of a negative chest X-ray, and high suspicion of FB aspiration, it is necessary to perform bronchoscopy [6,15].

The rigid bronchoscope has been very useful in the diagnosis and management of diseases of the airway; the first reports made in the last century already spoke of a success rate of up to 98.3% in the extraction of FBs from the airway [16].

Rigid bronchoscopy is considered the first diagnostic–therapeutic option for the obstruction of the airway by an FB [16]. Some of the benefits provided by the rigid bronchoscope are maintenance of the airway and ventilation while anesthesia is administered, a large working channel, and the availability of large forceps and other tools to remove FBs. Some disadvantages found are the need for a high level of training and the low availability in health care centers. Many experts consider rigid bronchoscopy as the gold standard in the diagnosis and management of the aspiration of an FB in children [8,16].

However, a review by Salih et al. [2] states that flexible fiber-optic bronchoscopy is considered as the gold standard procedure for the diagnosis and treatment of FBs as it provides direct visualization of airways where the FB is lodged. Flexible bronchoscopy is also safe, cost effective, and preferred by many pediatricians as it avoids the need for general anesthesia in comparison with rigid bronchoscopy

At present, there is evidence that shows the usefulness of the flexible bronchoscope in the extraction of FBs from the airway in both adults and children. Multiple series of cases have been reported (Table 2) which have shown high success rates [14,15,16,17,18,19]. 

Today, there are no prospective randomized trials to show which method is better^35^. In medical practice, one can be complementary to the other, so many experts recommend that, in the case of using the flexible bronchoscope, the bronchoscopist has at hand the rigid bronchoscope and is familiar with its use [14].

Complications associated with the removal of FBs are usually minimal, especially if the procedure is performed by expert personnel [14]. The mortality rate associated with the procedure varies from 0.13% to 2.0% [2,14].

Complications associated with bronchoscopy have been classified as minor and major [2]. Within the minor complications, trauma of the lips, teeth, tongue, epiglottis and larynx, minor hemorrhage, hypoxia, bradycardia, bronchospasm and mild laryngeal edema, fever, and subcutaneous emphysema have been observed [2,14,20].

It has been observed that the major complications that appear are usually associated with mortality. These include laryngospasm and severe bronchospasm, severe laryngeal edema requiring tracheotomy or reintubation, hypoxic brain damage, infections, atelectasis, pneumomediastinum, tracheal or bronchial laceration, perforation of the airway, failed bronchoscopy requiring a tracheotomy or thoracotomy, hemorrhage, pneumothorax, cardiac arrhythmias, and cardiac arrest [2,6,12].

Sometimes, as in the case of our patient, an additional tracheotomy may be performed in order to reduce risks, to facilitate the removal of the FB or to protect the airway [21]. In the specific case of our patient, it was necessary to perform a tracheostomy before the impaction of the FB in the subglottic space and after multiple unsuccessful extraction attempts. Some experts recommend the tracheostomy to ensure the airway in the presence of large FBs impacted in the subglottic space, and to maintain the tracheostomy cannula for approximately 5 days after extraction [21].

## 4. Conclusions

The aspiration of an FB is a serious clinical condition which occurs predominantly in the pediatric age group and, in many cases, has fatal outcomes. Timely diagnosis is the key to successful management. The history of an episode of choking associated with the presence of acute respiratory distress and subcutaneous emphysema should make the emergency doctor suspect airway obstruction due to an FB. Bronchoscopy is the gold standard in the management of the obstruction of the airway by an FB. In this sense, flexible bronchoscopy is an effective and highly useful tool for removing foreign bodies in pediatric patients, with low complication rates in expert hands.

## Figures and Tables

**Figure 1 medicina-58-00396-f001:**
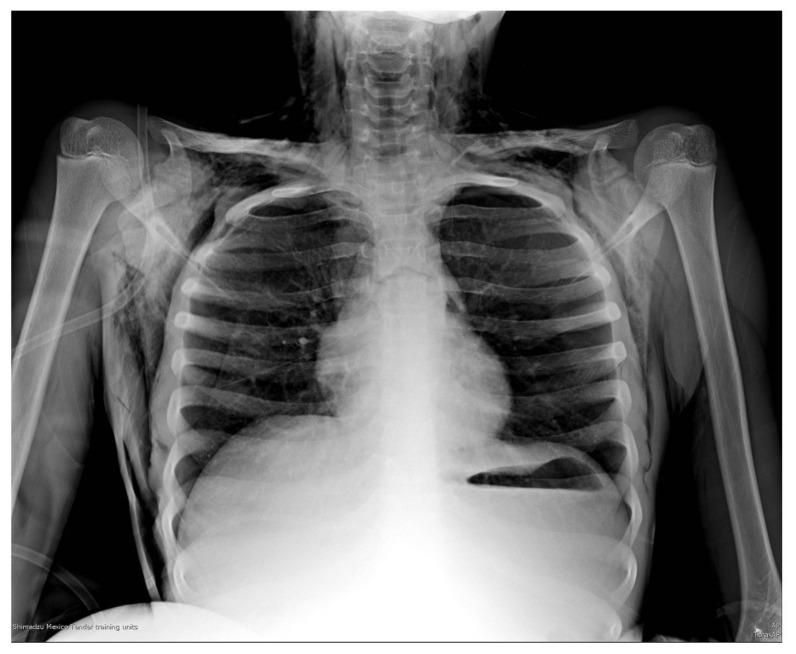
Plain chest radiograph in anteroposterior projection on admission of patient to the ED. A small pneumothorax can be observed in the upper right part of the left hemithorax of approximately 20%, as well as signs of pulmonary hyperinflation, with horizontalization of the costal arches; subcutaneous emphysema can also be observed in the neck and both hemithorax. Unable to visualize the FB.

**Figure 2 medicina-58-00396-f002:**
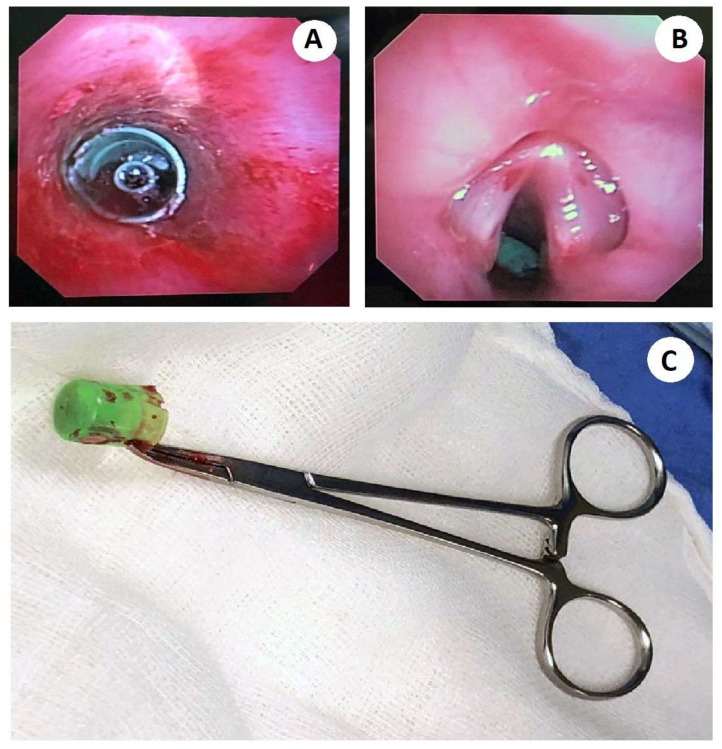
(**A**) Flexible bronchoscopy shows a tracheal FB with an evident near-total airway obstruction. The edematous trachea can be seen, with hyperemic mucosa and blood dotting. (**B**) Endoscopic view of the epiglottis, where the presence of edema is observed. The FB trapped in the subglottic space can be observed. (**C**) The photograph shows the foreign body extracted under flexible bronchoscopy turned out to be a plastic spray cap, cylindrical in appearance, with an approximate diameter of 1 cm, and a length of 2 cm.

**Table 1 medicina-58-00396-t001:** Types of bronchial obstruction that occur in the presence of a foreign body in the airway.

Type of Obstruction	Physiology
Type I or bypass valve	Partial obstruction of light in both phases of respiration with decreased aeration
Type II or check valve	Allows air flow during inspiration, but not during expiration.
Type III or stop valve	Air flow is not allowed either during inspiration or expiration, mainly in the event of a total obstruction, or the evolution of a type II obstruction
The type IV or ball valve	The FB is displaced during expiration, but is impacted again during inspiration

**Table 2 medicina-58-00396-t002:** Case series of the extraction of FBs using flexible bronchoscope.

Number of Bronchoscopies	Success Rate (%)	Reference
23	91.3%	[14]
24	100.0 %	[15]
28	100.0%	[16]
457	83.6%	[17]
938	91.3%	[18]
300	89.0%	[19]

## Data Availability

The datasets used and/or analyzed during the current study are available from the corresponding author on reasonable request.

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
