# Peer review of "Successful Management Foreign Body Aspiration Associated with Severe Respiratory Distress and Subcutaneous Emphysema: Case Report and Literature Review"

_medicina, 2022, doi:10.3390/medicina58030396_

Round 1
Reviewer 1 Report
The references (28) for a case report is too much; few older references can be omitted.
Author Response
dear reviewer 1
thanks for his contribution, the number of bibliographic citations was reduced according to his recommendation.

Reviewer 2 Report
Dear authors;
Firstly I appreciate your effort to this case. Please my comments and recommendations below;
- I believe that, your case about foreign body aspiration will contribute to literature certainly. Because FBA is a common and life-threatening condition. All pediatricians should suspect and exclude the foreign body aspiration possibility in every children with respiratory distress. So I think this issue always worth to review and publish as a reminder to clinicians.
- Please do not use abbreviations in abstract.
- Please review all abbreviations in the entire text and be cautious to use them at their first place in the manuscript (there is no explanation for FB, ER, PICU, CDC in their first place). Please check all abbreviations.
- Figures are appropriate.
- Please clarify how and why did you suspect foreign body aspiration in your patient in the case presentation section. Why did you decide to referee your patient to pediatric pulmonology suddenly. Please give detaile information about the FBa suspicion process.
- I’m not sure the number of authors in your references. According to me you could not write all author names in references and you should use ‘et al.’ after 3 or 6 authors names. Please check the author informations of the journal for your reference list.
- I want to share a study about pediatric foreign body aspiration with a large patient population from my country. I suggest you to provide information.
Aslan N, Yıldızdaş D, Özden Ö, Yöntem A, Horoz ÖÖ, Kılıç S. Evaluation of foreign body aspiration cases in our pediatric intensive care unit: Single-center experience. Turk Pediatri Ars. 2019 Mar 1;54(1):44-48. doi: 10.14744
Hope you can take my recommendations as a way to improve your study. I ’m willing to re-review your study after the required corrections made. Kind regards.
Author Response
dear reviewer 1
We appreciate your valuable comments to enrich the manuscript.
1. Thank you very much for your contributions.
2. Corrected abstract and removed abbreviations
3. The abbreviations were corrected in the manuscript and the missing ones were added.
4. Thank you very much for the contribution.
5. Since admission to the ER, the presence of a FBa was suspected. The choking episode, with subsequent respiratory distress, was witnessed by his classmates and reported to the emergency services. line 47-50.
Why did you decide to refer your patient to pediatric pulmonology suddenly. Please give detailed information about the FBa suspicion process.
Dear reviewer 2. Given the absence of a pediatric pulmonology service, and the impossibility of performing a bronchoscopy in our hospital, he was referred to the pediatric pulmonology service of a tertiary care hospital (our reference hospital), due to the suspicion of an FBa . line 70-72.
6. Dear reviewer 2, thank you for your suggestion, although the format of other journals suggests the use of "et al" for more than 3 authors, this journal does use the names of all authors.
7. I want to share a study about pediatric foreign body aspiration with a large patient population from my country. I suggest you to provide information.
Dear reviewer 2, thank you very much for the bibliographical suggestion, we have reviewed the suggested article and found it very interesting, it was attached as a bibliographical reference in our manuscript. It is bibliographic reference number 7.

Reviewer 3 Report
I have a few questions for the authors:
- The rigid bronchoscopy is very useful when the flexible fiber-optic bronchoscopy has failed to extract the foreign body. In the case presented, why was it not used? What were the arguments for choosing emergency tracheostomy over rigid bronchoscopy?
- What is the explanation for abundant blood secretion in the oral cavity?
- Can you confirm that, at least on the left, there is a small pneumothorax in the upper part of the hemithorax? it is not mentioned in the X-Ray chest description.
Author Response
dear reviewer 3
We appreciate your comments and suggestions to improve the article.
1. The rigid bronchoscopy is very useful when the flexible fiber-optic bronchoscopy has failed to extract the foreign body. In the case presented, why was it not used?
Dear reviewer 3 , The performance of flexible bronchoscopy under general anesthesia was considered as the appropriate diagnostic-therapeutic method (since a rigid bronchoscope was not available), being carried out two hours after their arrival (and five hours after the onset of the clinical picture). ). line 73-76.
2. What were the arguments for choosing emergency tracheostomy over rigid bronchoscopy?
Dear reviewer 3. Given the absence of a pediatric pulmonology service, and the impossibility of performing a bronchoscopy in our hospital, he was referred to the pediatric pulmonology service of a tertiary care hospital (our reference hospital), due to the suspicion of an FBa . The performance of flexible bronchoscopy under general anesthesia was considered as the appropriate diagnostic-therapeutic method (since a rigid bronchoscope was not available), being carried out two hours after their arrival (and five hours after the onset of the clinical picture). line 70-75.
3. What is the explanation for abundant blood secretion in the oral cavity?
Dear reviewer 3. The blood was of tracheal origin. line 78.
The bronchoscopy examination showed abundant blood secretions in the oral cavity "(most likely of tracheal origin)", as well as in the supraglottic space, with epiglottis in omega, cartilage without alterations. Glottic space with vocal cords without alterations
4. Can you confirm that, at least on the left, there is a small pneumothorax in the upper part of the hemithorax? it is not mentioned in the X-Ray chest description.
dear reviewer 3, it is correct there is a small pneumothorax, this information was added in the description of the x-ray. line 67-68. "a small pneumothorax located in the upper part of the left hemithorax, as well as subcutaneous emphysema in the neck and chest".

This manuscript is a resubmission of an earlier submission. The following is a list of the peer review reports and author responses from that submission.
Round 1
Reviewer 1 Report
The authors present the case of an adolescent with severe obstruction due to a foreign body and the presence of subcutaneous emphysema, a rare and severe condition in cases of obstruction. The novelty of the case is that the use of flexible bronchoscopy is proposed, which is not the procedure used for the definitive treatment of this pathology in children, in addition to indicating that the presence of subcutaneous emphysema can guide us towards its location. of the foreign body in the tracheal carina and this can help the emergency physician to definitive and rapid treatment for the patient. It also raises the use of tracheotomy, since there is not enough information on the use of this procedure in severe cases of obstruction. I suggest highlighting the importance of the use of the flexible bronchoscope in the conclusions and in the abstract, since it is only mentioned that the bronchoscope is the gold standard.
Reviewer 2 Report
- “At auscultation, the vesicular murmur was audible, although greatly diminished in its intensity at inspiration, it was not audible on exhalation.” Was echocardiography done?
- Full laboratory tests were performed, including arterial blood gases and simple chest X-ray were requested. What does the authors mean by “full laboratory tests” and “simple chest x-ray”? Please explain.
- Where is the lateral chest x-ray view?
- “However, it presented frank hyperinflation data, as well as subcutaneous emphysema in the neck and chest.” What was the reason of having subcutaneous emphysema in this child? Was it the complication of bronchoscopy?
- Correct the spelling.
- Why was CT-chest not done prior which is an important modality in localizing the foreign body?
- Authors state that “Rigid bronchoscopy is considered the first diagnostic-therapeutic option for obstruction of the airway by a FB.” Please cite this statement. A review by Salih AM “Salih AM. Airway foreign bodies: A critical review for a common pediatric emergency. World J Emerg Med. 2016; 7(1): 5.” States that flexible fiber-optic bronchoscopy is considered as a gold standard procedure of diagnosis and treatment for FB as it provides direct visualization of airways where the FB is lodged. Flexible bronchoscopy is also safe, cost-effective, and preferred by many pediatricians as it avoids the need for general anaesthesia in comparison with rigid bronchoscopy.
- What was the time duration between foreign body aspiration and bronchoscopic removal of the foreign body?
- Please correct the grammatical errors and typhos highlighted in the text.

Reviewer 3 Report
Authors present a case report of boy with foreign body aspiration and empyhsema. There is no new information in the manuscript and authors present a possible complication of FBA.